# Bio-Based Composites for Light Automotive Parts: Statistical Analysis of Mechanical Properties; Effect of Matrix and Alkali Treatment in Sisal Fibers

**DOI:** 10.3390/polym14173566

**Published:** 2022-08-29

**Authors:** Roberta Anastacia Palermo Fernandes, Pedro Henrique Poubel Mendonça da Silveira, Beatriz Cruz Bastos, Patricia Soares da Costa Pereira, Valdir Agustinho de Melo, Sergio Neves Monteiro, Neyda de La Caridad Om Tapanes, Daniele Cruz Bastos

**Affiliations:** 1Departamento de Materiais, State University of Rio de Janeiro, West Zone Campus—UERJ-ZO, Avenida, Manuel Caldeira de Alvarenga, 1203-Campo Grande, Rio de Janeiro 23070-200, Brazil; roberta_anas@hotmail.com (R.A.P.F.); patricia.soares.pereira@uerj.br (P.S.d.C.P.); valdir.melo@gmail.com (V.A.d.M.); neyda.tapanes@uerj.br (N.d.L.C.O.T.); daniele.bastos@uerj.br (D.C.B.); 2Department of Materials Science, Military Institute of Engineering—IME, Praça General Tibúrcio, 80, Urca, Rio de Janeiro 22290-270, Brazil; snevesmonteiro@gmail.com; 3Federal Institute of Education, Science and Technology of Rio de Janeiro—IFRJ, Rua Lúcio Tavares, 1045-Centro, Nilópolis 26530-060, Brazil; beatrizbastos026@gmail.com

**Keywords:** polymeric composites, sisal fiber, factorial experimental design methodology, polypropylene, automotive, alkali treatment

## Abstract

Composites based on virgin and recycled polypropylene (PP and rPP) reinforced with 15 wt% sisal fibers, with and without alkali treatment, were prepared by compression molding in a mat composed of a three-layer sandwich structure. The sisal was characterized by Fourier-transform infrared spectroscopy (FTIR) and X-ray diffraction (XRD). The composites were characterized according to physical and mechanical properties. Additionally, a factorial experimental design was used to statistically evaluate the mechanical properties of the composite. The FTIR and XRD indicated the partial removal of amorphous materials from the surface of the sisal after alkali treatment. The composites’ density results varied from 0.892 to 0.927 g·cm^−3^, which was in the desirable range for producing lightweight automotive components. A slight decrease in the hardness of the pure rPP and rPP composites in relation to the PP was observed. The water absorption was higher in rPP composites, regardless of the chemical treatment. Moreover, the impact resistance of PP and its composites was higher than the values for rPP. Statistical analysis showed that the alkali treatment was a significant factor for the hardness of the rPP and PP composites, and that the addition of the sisal layer was relevant to improve the impact resistance of the composites.

## 1. Introduction

In the current linear economic model, accumulation of waste population and consumption result in increasing waste and depletion of natural resources, exacerbating the problems of climate change problem and resource scarcity. In reaction, the circular economy concept has gained importance, aiming to minimize the environmental impacts by reducing consumption of energy, water, chemicals and materials so wastes lowering emissions of greenhouse gases [1,2].

In order to minimize environmental impacts, many industries, such as the automotive industry, have developed lightweight materials as components in the construction of automobiles. Lightweight materials are advantageous because they require a smaller amount of materials in their manufacture, reducing cost and processing time, as well as improving the energy efficiency of vehicles due to the reduction in total weight [3,4]. The most common fillers used in lightweight materials in the automotive industry are glass fibers, talcum and calcium carbonate. These reinforcements are used to provide strength and stiffness, and control manufacturing cost [5,6,7]. However, there is a trend to replace these reinforcements with materials with lower density and cost, such as natural fibers [8].

Natural fibers have been investigated for several applications as reinforcing fillers for aeronautics, automobiles, construction materials, and ballistic armors [9,10,11,12,13,14,15]. In the automotive industry, fibers such as jute [16], flax [17], hemp [18], kenaf [19] and many others are used as reinforcement in polymeric matrix composites (PMCs) to improve mechanical properties. The effect of natural fiber loading on physical and mechanical characteristics of polypropylene (PP)-based composites is also reported in the literature, using sisal, ramie and sisal/ramie fibers [20], as well as corn fibers [8], leaves of *Yucca aloifolia* L. [21] short flax and pine fibers [22]. However, biocompatibility and appropriate processing windows are two major problems faced, since the properties of composites change depending on each component material. Moreover, it is important to develop bio-based composites with suitable processing techniques to attain superior physical, mechanical and thermal properties. The surface treatment of the fibers allows them to have greater adhesion to the polymer matrix by removing components such as lignin from their surface [23,24].

Different chemical treatments can be adopted to improve the mechanical and surface properties of natural fiber-reinforced composites due to the fundamental problem associated with the hydrophilic nature of fibers. Natural fibers are comparatively more moisture absorbent and exhibit lower strength than synthetic fibers. Researchers are aiming to develop pretreatment processes to improve the compatibility of natural fibers with polymer matrices and to enhance the mechanical characteristics of the composites. The alkali treatment approach is one of the least complex, economical and powerful techniques applied for enhancing the adhesion of natural fibers to polymeric matrix [25,26,27].

Sisal fiber (*Agave sisalana*) is among the most-used natural fibers in the world. The sisal plant is easy to cultivate, can be grown in several regions and has short production times [28,29]. The fiber has low density and high resistance, and is used to make artifacts such as ropes, carpets, rugs, fabrics, as well as composite materials. An application of sisal fibers in automotive components may be an economically viable alternative, since the low cost of the fibers can reduce the manufacturing value of parts produced with thermoplastic polymers [30,31]. The method for preparation of PP/Sisal composites in this work was compression molding, which was chosen by taking th advantage of the sisal fibers to be in a fabric form. During hot compression of fibers, the inner part of the fibers does not melt, remains highly oriented and acts as reinforcement [32].

The goal of the present work was to obtain bio-based composites based on virgin and recycled polypropylene (PP and rPP), as well as sisal fabric, with and without alkali treatment. The formulations were prepared by compression molding in a mat with a three-layer sandwich structure: Matrix/Filler/Matrix (PP–Sisal–PP; PP–Sisal NaOH-PP; rPP–Sisal–rPP, and rPP–Sisal NaOH-rPP). The sisal was characterized by Fourier-transform infrared spectroscopy (FTIR) and X-ray diffraction (XRD). The processed formulations were characterized according to physical and mechanical properties (density, water absorption, hardness, and Izod impact). A factorial experimental design was used to statistically evaluate the mechanical behavior of the bio-based composites.

## 2. Materials and Methods

### 2.1. Materials

The virgin polypropylene (PP) used in this work was purchased from Braskem S/A (molecular weight: 470,000 g/mol, density: 0.905 g·cm^−3^ and melt flox index: 3.5 g/10 min at 230 °C/2.16 kg). Black recycled polypropylene (rPP) was donated by Polialbino Termplásticos (São Paulo, Brazil) and sisal fabric was purchased from a conventional market located in Rio de Janeiro, Brazil. The sisal fabric was cut into specimens of 15 cm^2^.

### 2.2. Alkali Treatment of Sisal and Bio-Based Composites Processing


The alkali solution was prepared with a concentration of NaOH 10% (*w*/*v*) with distilled water. At first, the clean and dried sisal specimens were soaked in an alkali solution at room temperature for 24 h with constant stirring. The specimens were then washed several times with distilled water to neutralize the remaining NaOH in the sisal, and then dried at 100 °C in an oven for 24 h before processing. The unprocessed sisal was denoted “Sisal”, while the alkali-treated sisal was denoted “Sisal NaOH”. The bio-based composites were prepared by compression molding. A mat with a three-layer sandwich structure (Matrix/Filler/Matrix; see Figure 1), was hot pressed at 190 °C and 6 tons for 5 min. Next, 15 g of PP was added to form the first layer, followed by the addition of one layer of sisal fabric, with approximately 5 g, and completing the sandwich, another layer of 15 g of PP, forming a composite of approximately 15 wt% reinforcement.

The thickness of composite veneer panel was controlled by two steel bars of 1.36 mm thickness between the hot press boards. After hot pressing, the composite veneer panel was cold compressed to room temperature for 4 min at 6 tons. The processed formulations were named PP; rPP; PP/Sisal; PP/Sisal NaOH; rPP/Sisal and rPP/Sisal NaOH.

### 2.3. Characterization

#### 2.3.1. Fourier Transform Infrared Spectroscopy (FTIR)


Fourier-transform infrared spectra (FTIR) were acquired using a Nicolet 6700 FTIR spectrometer (Thermo Fisher Scientific). The unprocessed sisal (Sisal) and NaOH-treated sisal (Sisal NaOH) were mounted on an attenuated total reflectance (ATR) accessory equipped with ZnSe crystal prior to scanning. The spectra were obtained with an accumulation of 120 scans and resolution of 4.182 cm^−1^.

#### 2.3.2. X-ray Diffraction (XRD)

X-ray diffraction (XRD) analysis of Sisal and Sisal NaOH were performed with a Shimadzu XRD-6000 diffractometer (Tokyo, Japan) with CuK 
α
 radiation, power of 1200 watts (40 kV × 30 mA), with a 2
θ
 sweep from 5° to 40°. The crystallinity index (CI) was calculated based on the methodology presented by Segal et al. [33], in which the intensity of the amorphous phase of the fiber and the crystalline phase, referring to the 002 plane, is used. The calculation of the crystallinity index is presented as follows:
(1)
$$\mathrm{CI}=\frac{I_{002}-I_{amorphous}}{I_{002}}\cdot 100\%$$


#### 2.3.3. Water Absorption

Water absorption test (%WA) of the polypropylene composites was performed according to ASTM D-570 [34]. Three samples with dimensions 20 × 20 × 0.2 mm were inserted into a recipient with distilled water at room temperature, in which they were immersed for different times (2, 24, 168 and 336 h). After the immersion time, the excess moisture was removed from the surface of the samples, and then weight measurements were taken. From Equation (Equation 2), described below, the absorption was calculated from the difference in mass of the dry and wet samples.

(2)
$$\%\mathrm{WA}=\frac{w_{\mathrm{final}}-w_{\mathrm{initial}}}{w_{\mathrm{initial}}}\cdot 100\%$$


#### 2.3.4. Density Measurements

Density analyses were performed according to ASTM D792 [35]. A Gehaka DSL910 densimeter (São Paulo, Brazil) was used at room temperature, in which five samples from each group were characterized and for density determination.

#### 2.3.5. Shore-D Hardness

Shore D hardness tests were performed according to ASTM D2240-05 [36]. The Shore D Durometer (Type GS 702) provided the Shore D hardness value of the material analyzed. For each sample, PP, rPP and bio-based composite, the highest and lowest values were excluded; thus, we calculated the arithmetic mean of the five determinations.

#### 2.3.6. Izod Impact Test

The Izod impact strength test of the processed specimens (PP, rPP and bio-based composites) was conducted according to ASTM D-256 [37] using a universal pendulum impact tester. The samples were rigidly mounted in the vertical position and struck by a pendulum with a force of 7.5 J at the center of the samples.

#### 2.3.7. Statistical Analysis

This study uses multiple linear regression to obtain mathematical models that relate the properties of virgin PP and rPP bio-based composites with the preparation conditions, in order to optimize the mechanical performance of the bio-based composite. Analysis of variance (ANOVA) was performed to evaluate the models, and the response surface methodology (RSM) was applied for optimization. The number of sisal layers (SS) and alkali solution concentration for treatment of the composite (AT) were used as independent variables, also called factors. The minimum and maximum levels of SS and AT were considered 0–1 and 0–15 % *w*/*v*, respectively. We also used a dummy variable for the PP matrix, of M = 1 for virgin PP and M = 2 for recycled PP.

## 3. Results and Discussion

### 3.1. Fourier Transform Infrared Spectroscopy (FTIR)

The ATR-FTIR spectra of Sisal and Sisal NaOH are shown in Figure 2. It can be observed that the effect of alkali treatment was the removal of excess amorphous constituents (hemicellulose and lignin) in comparison with the untreated specimens. NaOH treatment, or alkali treatment, dissolves the amorphous constituents by removing the hydroxyl bond, which can improve surface roughness by creating strong interfacial bonding between the fibers and the matrix [38].

The sisal showed a wide band at 3296 cm^−1^, related to the O-H stretching vibration of the hydroxyl groups present in the cellulose molecules. At 2914 and 2796 cm^−1^ were bands related to C-H stretching vibration of alkyl groups in aliphatic bonds of cellulose, lignin and hemicelluloses [39]. The band at 1737 cm^−1^ was attributed to acetyl and ester groups of hemicelluloses and aromatic components of lignin. The band at 1642 cm^−1^ was related to the O-H bending of the absorbed water. At 1346 cm^−1^, a band related to the stretching of the C=O bond of hemicellulose components was displayed. At 1028 cm^−1^, an absorption band related to the stretching of C-O and OH bonds owed to lignin and cellulose components occurred [40,41,42,43,44]. The alkali treatment resulted in a reduction in components such as lignin and hemicellulose in the structure of the sisal. It is observed that the removal of these components occurred due to the reduction in intensity of the bands at 1737, 1646, 1347 and 1028 cm^−1^, which was associated with bands directly associated with lignin and hemicellulose.

Table 1 summarizes the relationship between absorption bands, functional groups and fiber and their components.

### 3.2. X-ray Diffraction (XRD)

The diffractograms of Sisal and Sisal NaOH are shown in Figure 3.

The diffractograms of the sisal show a pattern quite similar to the four peaks characteristic of native cellulose (cellulose I). These peaks are located at 2
θ
 = 10.41°, 16.30° and 29.50°, which are the positions of the crystallographic planes (101), (002) and (040), respectively [45,46,47]. Using the calculation of Segal et al. [33], a significant increase in the crystallinity index of Sisal NaOH fiber was observed compared to the untreated sisal fabric, in which an increase in CI from 44.76% of the untreated fiber to 73.18% of the NaOH-treated fiber. This indicates the partial removal of amorphous materials from the surface of sisal fibers, corroborating the FTIR results.

Several papers have reported the increase in crystallinity after NaOH treatment of natural fibers. The removal of lignin and hemicellulose results in an increase in crystallinity. In application in composites, this removal increases interfacial adhesion with the polymer matrix, resulting in improved mechanical properties [48,49]. This is evidenced in recent work, in which the treatment of fibers such as *Perotis indica* [50], *Hibiscus vitifolius* [51], *Heteropsis flexuosa* [52], *Grewia Flavescens* [23] and Henequen [53] resulted in increased mechanical properties and thermal resistance of polymer matrix composites.

### 3.3. Water Absorption

Figure 4 and Table 2 shows the water absorption results. A trend of increasing mass is evident after 2 h. After that, the absorption values of all groups, except PP, decrease to values close to those obtained for the 24 h period of the test. Pure PP absorbed the most water within 2 h of testing, while rPP absorbed the most water with increasing testing time. The addition of sisal fabric promoted an increase in absorption, which was observed for the samples PP/Sisal, rPP/Sisal, PP/Sisal NaOH and rPP/Sisal NaOH. This increase became more pronounced as a function of increasing test time. The PP/Sisal NaOH sample exhibited lower absorption values than the other bio-based composites.

The high liquid absorption is related to the hydrophilic nature of the sisal fiber [54]. The high absorption values of the composites indicate that the fiber absorbed a significant amount of water throughout the test. Pure PP showed the lowest water absorption values, disclosing a weight increase under 1% throughout the test, reinforcing the fact that the fiber absorbed most of the liquid in the composite samples.

The rPP also showed low liquid absorption in the test, however its gain was higher than the PP sample. This reflects in the composites, in which the recycled matrix with addition of 15 wt% of sisal fabric resulted in the highest absorption results: 7.914 and 9.051% for the composites rPP Sisal and rPP Sisal NaOH, respectively. The recycling of polypropylene may have affected the performance of the bio-composite, because during the recycling process, the polymer is subjected to thermal, mechanical and chemical treatments, which can impair the properties [55,56].

### 3.4. Density, Hardness and Impact Strenght

Table 3 summarizes the results obtained for density, hardness and Izod impact strength for the bio-based composites.

The density results ranged from 0.89 to 0.95 g·cm^−3^. The results state that the bio-based composites did not change significantly in density, which is important for application in lightweight automotive components. The density values of the PP and rPP samples were in agreement with literature data [57]. The addition of sisal caused an increase in the density of the composites. This increase occurred due to the density of the sisal (1.4 g·cm^−3^) being higher than the density of polypropylene (0.9 g·cm^−3^) [58,59].

From the hardness results in Table 3, it can clearly be seen that the recycled polypropylene hardness is lower than that of virgin polypropylene. This difference in polymer hardness in the two conditions reflects directly on the hardness of the composites. The PP matrix bio-composites composites (PP/Sisal and PP/Sisal NaOH) exhibit a small increase in the hardness value, attenuated by the standard deviation. The rPP/Sisal composite obtained a small increase in hardness compared to rPP (from 58 to 59.33 Shore D); however, the values were lower than those of the PP and PP matrix bio-based composites. The PP/Sisal NaOH bio-composite presented the lowest hardness among all groups (57 Shore D). The recycled polymer did not exhibit good adhesion with the treated fiber, which is also reflected in the impact strength of the bio-based composites.

By considering the impact strength results in Table 3, the impact strength decreased when rPP was used as a matrix, which was expected to some extent based on the composition of the recycled matrix. Another contributing factor could be undesirable agglomeration [4]. In the recycled matrix, the impact resistance values decreased, showing filler–matrix incompatibility, which culminated in greater water absorption and lower mechanical strength in comparison with virgin matrix and its bio-based composites [60,61]. The groups with rPP matrix (rPP, rPP/Sisal and rPP/Sisal NaOH) exhibited low flexural strength values, because there was no interfacial adhesion between the polymeric matrix and the sisal fiber, either untreated or treated. Virgin PP showed higher values than its bio-based composites due to the hydrophilic character of natural fibers. Added to the hydrophobicity of polypropylene, the composites showed poor adhesion between the matrix and the reinforcement phase. The alkaline treatment also did not prove to be effective in the adhesion of the fibers to the polymer, in which the reduction of the impact resistance is more accentuated in the rPP bio-based composites. The use of virgin polypropylene in the processing resulted in composites with higher impact strength. The PP/Sisal and PP/Sisal NaOH bio-based composites presented impact strengths of 9.22 and 9.14 MPa, respectively, higher results than the groups with rPP matrix.

### 3.5. Statistical Analysis

The experimental matrix performed from the measured responses for hardness (H), Izod impact strength (I) and density (D) were represented in Table 4. The analysis of variance (ANOVA) calculations for the three tests discussed in the previous topic, are next in Table 5, Table 6 and Table 7. ANOVA was used to detect the factors and their interactions that significantly influence the mechanical performance of the composites. The second-degree interactions considered in the study are M-AT, M-SS and AT-SSL.

The *p*-values for the factors (M, AT and SS) and interactions (M-AT, M-SS and AT-SS) shown in Table 4, Table 5 and Table 6 indicate the influence on the response variables (H, I and D). As the level of confidence was considered to be 95%, then if *p*-value of the factor or the interaction was lower than or equal to the risk degree (0.05 or 5%) there was a significant correlation between the response variables and the factor, while *p*-values higher than 0.05, indicated the absence of correlation. The results illustrated in Table 4 show that the factor AT and the interaction M-AT had statistical significance on the response variable H, confirming the relevant influence of the alkali treatment on the hardness of the composite. In contrast, the regression results showed that the impact resistance response was not influenced by the AT factor (Table 6). It can also be observed that the addition of the sisal layer to the rPP or PP composites was relevant for the impact resistance, thus recommending its use. The M variable represents the most significant factor in all analyses, a result corroborated in the Pareto diagrams, Figure 5 and Figure 6. The reference line on the graphs indicates which effects are significant. In this study, Lenth’s method [62] was used to draw the reference line. Terms with effects to the right of the line represent significant parameters.

The adjusted R-Sq (adj) values for the regression models of the H, I and D responses are 87.31%, 71.20% and 70.60%, respectively. These coefficients presented levels above 70%, which implies that the models had good predictability. Residual analysis was performed to check for the assumptions of ANOVA and validate the regression models. The model’s H, I and D responses were determined by Equations (3)–(5).

(3)
H=68.0−5.0·M+0.156·AT−0.156·M−AT±0.577


(4)
I=34.514−13.017·M−22.896·SS+10.618·M−SS±2.639


(5)
D=0.868+0.027M−0.002·AT+0.035·SS±0.13


## 4. Summary and Conclusions

In this work, composites were obtained via compression molding, varying the type of matrix, virgin polypropylene (PP) and recycled polypropylene (PPr) and treatment of sisal fabric used as filler. From the results discussed, the following conclusions were drawn

The FTIR analysis indicated that the alkali treatment removed most of the amorphous materials, such as hemicellulose, and lignin from the surface of the sisal fibers.X-ray diffraction analysis of the sisal showed that the alkali treatment promoted a significant increase in its crystallinity, in which the crystallinity index increased from 44.76% to 73.18%, corroborating the FTIR analyses.The density of the materials ranged from 0.89 to 0.95 g·cm^−3^, showing the possibility of using these bio-based composites in light automotive parts.The rPP and rPP composites had lower hardness than those of virgin PP.Water absorption tests revealed that the composites with the rPP matrix showed higher liquid absorption, due to the lack of sisal–matrix interaction and the hydrophilic character of the fiber, besides the presence of the fibers that had hydrophilic behavior. All groups except PP presented the highest liquid absorption in the 168 h test.The composites with the PP matrix displayed higher-impact strength values than the composites with the rPP matrix, corroborating the results of water absorption and hardness.Statistical analysis revealed that the type of bio-based composite matrix was the most significant variable. The regression model and the Pareto diagrams showed that the alkali treatment was a significant factor for the hardness of rPP and PP composites, and that the addition of a sisal layer was relevant to improve the impact resistance of the composites. For the fabrication of internal components of automobiles, the matrix of virgin PP and the alkali treatment of the fiber are indicated.

## Figures and Tables

**Figure 1 polymers-14-03566-f001:**
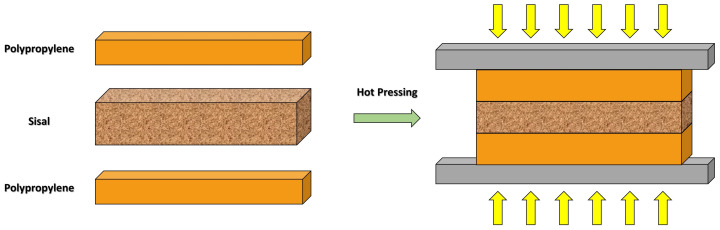
Scheme of the mat with a three-layer sandwich structure for composite preparation by compression molding.

**Figure 2 polymers-14-03566-f002:**
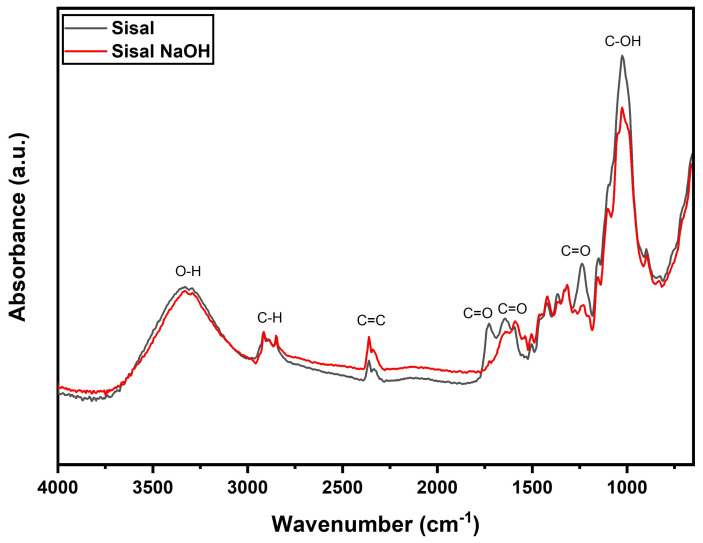
Overlay of the ATR-FTIR spectra of sisal fibers.

**Figure 3 polymers-14-03566-f003:**
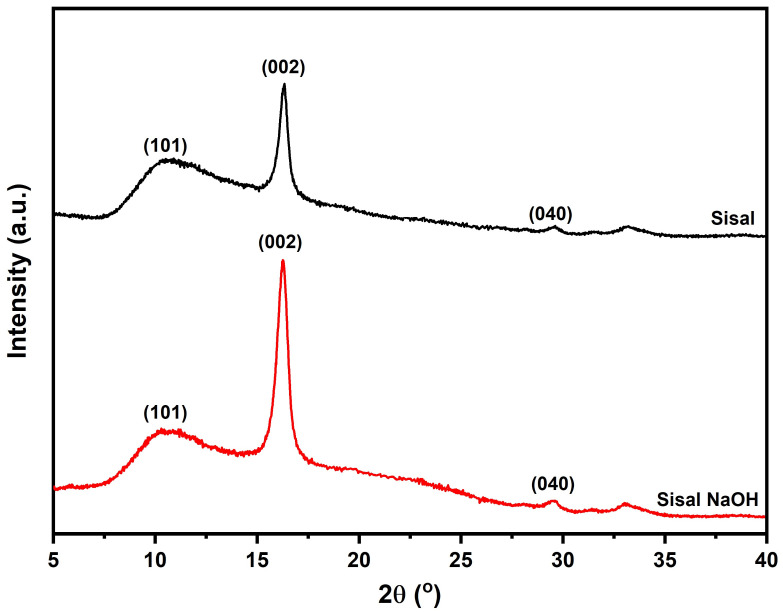
X-ray diffractogram of Sisal and Sisal NaOH.

**Figure 4 polymers-14-03566-f004:**
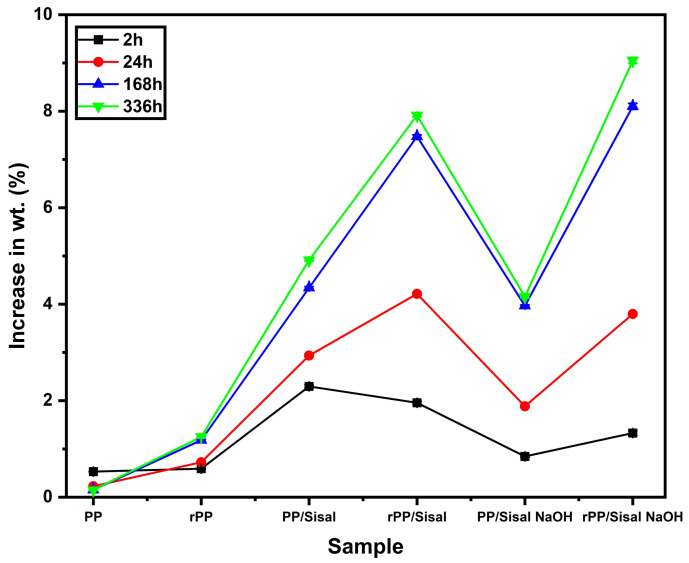
Water absorption results for PP, rPP and bio-based composites.

**Figure 5 polymers-14-03566-f005:**
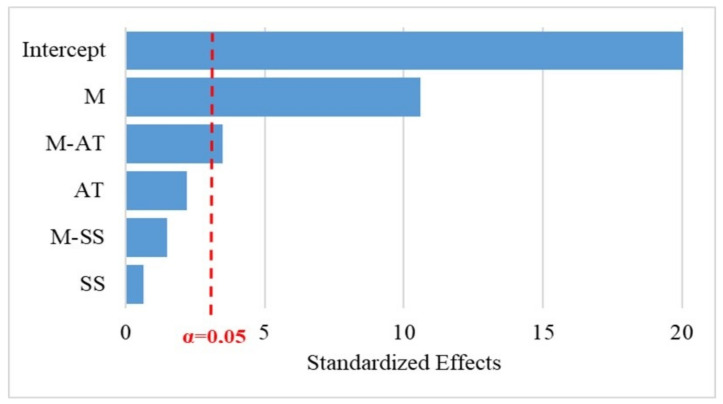
Standardized Pareto Chart for Shore D Hardness.

**Figure 6 polymers-14-03566-f006:**
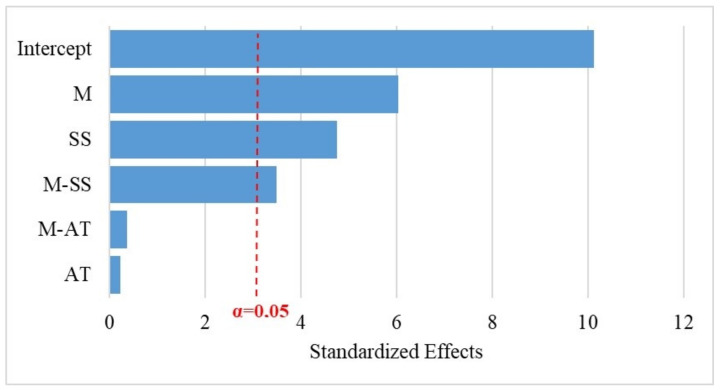
Standardized Pareto Chart for for Izod impact strength.

**Table 1 polymers-14-03566-t001:** Bands with vibrational modes assigned to the FTIR spectrum of sisal.

Absorption Band (cm^−1^)	Functional Group	Fiber Component
3296	O - H stretching	Cellulose
2914	C - H stretching	Cellulose
2796	C - H stretching	Hemicellulose and Lignin
2357	C = C stretching	Wax
1737	C = O stretching	Hemicellulose and Lignin
1642	OH bending	Lignin
1346	C = O stretching	Hemicellulose
1028	C - OH stretching	Lignin and Cellulose

**Table 2 polymers-14-03566-t002:** Water absorption results of processed formulations.

	Increase in wt. (%)			
Sample	Immersion Time			
	2 h	24 h	168 h	336 h
PP	0.531 ± 0.008	0.227 ± 0.008	0.152 ± 0.009	0.152 ± 0.008
rPP	0.592 ± 0.017	0.723 ± 0.017	1.183 ± 0.018	1.249 ± 0.018
PP/Sisal	2.297 ± 0.031	2.935 ± 0.028	4.339 ± 0.025	4.914 ± 0.028
rPP/Sisal	1.958 ± 0.031	4.214 ± 0.031	7.477 ± 0.032	7.914 ± 0.031
PP/Sisal NaOH	0.847 ± 0.017	1.885 ± 0.019	3.966 ± 0.017	4.156 ± 0.019
rPP/Sisal NaOH	1.329 ± 0.050	3.797 ± 0.057	8.101 ± 0.060	9.051 ± 0.057

**Table 3 polymers-14-03566-t003:** Density, Shore D hardness and Izod Impact of processed formulations.

Sample	Density (g·cm^−3^)	Shore D Hardness	Izod Impact Strength (MPa)
PP	0.89 ± 0.02	63.00 ± 0.00	21.49 ± 0.25
rPP	0.92 ± 0.01	58.00 ± 0.58	8.48 ± 0.08
PP/Sisal	0.93 ± 0.02	63.33 ± 1.00	9.22 ± 0.02
rPP/Sisal	0.95 ± 0.01	59.33 ± 1.00	6.82 ± 0.06
PP/Sisal NaOH	0.90 ± 0.02	63.33 ± 1.00	9.14 ± 0.07
rPP/Sisal NaOH	0.93 ± 0.01	57.00 ± 1.00	5.63 ± 0.05

**Table 4 polymers-14-03566-t004:** Experimental design matrix and the corresponding values of response variables.

Experiment	M	AT	SS	H	I	D
1	1	0	0	63	0.893	18.614
2	1	0	0	63	0.875	17.853
3	1	0	0	63	0.907	28.025
4	1	0	1	63	0.955	9.641
5	1	0	1	64	0.911	8.788
6	1	0	1	63	0.934	9.226
7	1	15	1	63	0.891	7.911
8	1	15	1	63	0.921	8.557
9	1	15	1	64	0.891	10.956
10	2	0	0	58	0.932	10.426
11	2	0	0	58	0.921	6.620
12	2	0	0	58	0.923	8.396
13	2	0	1	60	0.956	8.004
14	2	0	1	59	0.957	5.305
15	2	0	1	59	0.950	7.150
16	2	15	1	58	0.931	6.389
17	2	15	1	57	0.918	6.228
18	2	15	1	56	0.932	4.267

**Table 5 polymers-14-03566-t005:** ANOVA of factorial design for density measurements (D).

Source of Variation	Degree of Freedom (D.F)	Sum of Squares (SQ)	Means Squares (MQ)	F	Significance of F
Regression	3	0.0076	0.0025	14.6094	$1.36\times 10^{-4}$
Residue	14	0.0024	0.0002		
Total	17	0.0100			
**Term**	**Coefficients**	**Stand. Error**	**Stat t**		**Value-** * **p** *
intercept	0.8682	0.0107	80.7936		$4.31\times 10^{-20}$
M	0.0269	0.0062	4.3342		$6.87\times 10^{-4}$
AT	−0.0020	0.0005	−3.9264		$1.52\times 10^{-3}$
SS	0.0353	0.0076	4.6502		$3.75\times 10^{-4}$
S = 0.0132					
R-sq = 75.79%					
R-sq (adj) = 70.60%					

**Table 6 polymers-14-03566-t006:** ANOVA of factorial design for Shore D hardness (H).

Source of Variation	Degree of Freedom (D.F)	Sum of Squares (SQ)	Means Squares (MQ)	F	Significance of F
Regression	6	126	21	75.6	$2.6797\times 10^{-8}$
Residue	12	4	0.3333		
Total	18	130			
**Term**	**Coefficients**	**Stand. Error**	**Stat t**		**Value-** * **p** *
Intercept	68	0.7454	91.2316		$2.01\times 10^{-18}$
M	−5	0.4714	−10.6066		$1.889\times 10^{-7}$
AT	0.1556	0.0703	2.2136		0.0469
SS	−0.6667	1.0541	−0.6325		0.5389
M-AT	−0.1556	0.0444	−3.5		0.0044
M-SS	1	0.6667	1.5		0.1595
S = 0.5775					
R-sq = 96.92%					
R-sq (adj) = 87.31%					

**Table 7 polymers-14-03566-t007:** ANOVA of factorial design for Izod Impact Resistance (I).

Source of Variation	Degree of Freedom (D.F)	Sum of Squares (SQ)	Means Squares (MQ)	F	Significance of F
Regression	6	494.8954	82.4826	14.2137	0.00013
Residue	12	83.5634	6,9636		
Total	18	578,4588			
**Term**	**Coefficients**	**Stand. Error**	**Stat t**		**Value-** * **p** *
intercept	34.5137	3.4068	10.1309		$3.11\times 10^{-7}$
M	−13.0166	2.1546	−6.0413		$5.84\times 10^{-5}$
AT	0.0692	0.3212	0.2154		0.8330
SS	−22.8964	4.8179	−4.7524		0.0005
M-AT	−0.0743	0.2031	−0.3659		0.7208
M-SS	10.6178	3.0471	3.4846		0.0045
S = 2.6388					
R-sq = 85.55%					
R-sq (adj) = 71.20%					

## Data Availability

The data presented in this study are available on request from the corresponding author.

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
