# Peer review of "Bio-Based Composites for Light Automotive Parts: Statistical Analysis of Mechanical Properties; Effect of Matrix and Alkali Treatment in Sisal Fibers"

_polymers, 2022, doi:10.3390/polym14173566_

Round 1

Reviewer 1 Report

The development of polymer composite materials with natural fibers due to their environmental friendliness and low cost is an important area of researches. Replacing the usual glass and carbon fillers with natural fibers often leads to cheaper products. The work complies with the requirements of Polymers journal. However, there are a number of important comment, according to which the authors should make changes to the manuscript:

1. In the Abstract section, the rPP composites are written twice (line 9).

2. In the Introduction section the authors should give specific examples of the use of fillers for polypropylene and how fillers affect their physical and mechanical properties. For example, doi: 10.1016/j.compositesb.2021.109121 etc.

3. In the Introduction section it should be indicated which methods for laying out preforms exist (https://doi.org/10.3390/polym14010087), and it should be explained why the authors chose this particular method. Otherwise, the Introduction is more like an Abstract. Authors should review more articles and expand the Introduction.

4. In the Introduction section, it is necessary to explain in detail why the sisal fiber is treated with alkali (lines 42-43). What are the known physical and chemical methods of processing natural fibers?

5. In the Materials section, the molecular weight of polypropylene should be indicated.

6. Why does PP/Sisal NaOH composite have less water absorption than PP/Sisal composite (Figure 4)? Give an explanation.

7. What causes a significant reduction in the impact strength of polypropylene filled with sisal (table 3)? Give an explanation.

Author Response

The response to reviewers is attached.

Reviewer 2 Report

The article concerns research on composites based on synthetic and natural fibers, subjected to, i.a., alkaline treatment. Selected physical and mechanical properties of several combinations of composites were tested. To improve the quality of the manuscript, consider the following corrections:

The introduction focuses only on fillers, mostly natural fibers. Nothing has been written about the matrix, e.g. PP fibers, what is the structure of these composites and how they are produced. The article concerns the production process, so in the introduction it would be necessary to write something about it, how other composites are produced, found in literature or in practice.

Line 55: The introduction should not include details such as "with hot pressing of 6 tons at 190 ° C for 5 minutes" and sample markings (PP-Sisal-PP etc.). Information should be provided on what materials the composite is to consist of and that it is compressed under high temperature conditions. Details should be included in the "Materials and Methods" part.

Line 95: The last letter of the word "test" is missing

Table 2. The title of the table is inappropriate because the results apply to all tested composites, not only PP / Sisal

Figure 4 and Table 2: Why was there an increase in weight at point 0 when the samples were immersed in water? It means that samples dried to constant weight were not tested, only samples with air-dry mass? If so, please write it in the methodology.

Figure 4 and Table 2: Did the alkaline treatment of the composites affect the water absorption results? This should be commented on in the analysis of the results.

Figure 4 and Table 2: There is no deviation in the results. More than one sample was tested.

Table 3. What do the deviations in the results mean? Is it a standard deviation? This should be written.

Author Response

The response to reviewers is attached.

Round 2

Reviewer 1 Report

The authors took into account most of the comments

Reviewer 2 Report

The quality of the manuscript has improved after taking most of the comments into account.